# Attributes of Culture Bacteria as Influenced by Ingredients That Help Treat Leaky Gut

**DOI:** 10.3390/microorganisms11040893

**Published:** 2023-03-30

**Authors:** Ricardo S. Aleman, David Paz, Roberto Cedillos, Miguel Tabora, Douglas W. Olson, Kayanush Aryana

**Affiliations:** 1School of Nutrition and Food Sciences, Louisiana State University Agricultural Center, Baton Rouge, LA 70803, USA; 2Faculty of Technological Sciences, Universidad Nacional de Agricultura Road to Dulce Nombre de Culmí, Km 215, Barrio El Espino, Catacamas 16201, Olancho, Honduras

**Keywords:** yogurt starter culture, functional ingredients, acid tolerance, bile tolerance, protease activity, growth characteristics

## Abstract

Consumers are becoming aware of functional ingredients such as medicinal herbs, polyphenols, mushrooms, amino acids, proteins, and probiotics more than ever before. Like yogurt and its probiotics, L-glutamine, quercetin, slippery elm bark, marshmallow root, N-acetyl-D-glucosamine, licorice root, maitake mushrooms, and zinc orotate have demonstrated health benefits through gut microbiota. The impact of these ingredients on yogurt starter culture bacteria characteristics is not well known. The objective of this study was to determine the influence of these ingredients on the probiotic characteristics, tolerance to gastric juices and lysozyme, protease activity, and viability of *Streptococcus thermophilus* STI-06 and *Lactobacillus bulgaricus* LB-12. Acid tolerance was determined at 0, 30, 60, 90, and 120 min of incubation, whereas bile tolerance was analyzed at 0, 4, and 8 h. The microbial growth was determined at 0, 2, 4, 6, 8, 10, 12, 14, and 16 h of incubation, while protease activity was evaluated at 0, 12, and 24 h. The application of marshmallow root, licorice root, and slippery elm bark improved bile tolerance and acid tolerance of *S. thermophilus*. These ingredients did not impact the bile tolerance, acid tolerance, and simulated gastric juice tolerance characteristics of *L. bulgaricus* over 8 h and 120 min (respectively) of incubation. Similarly, the growth of *S. thermophilus* and *L. bulgaricus* was not affected by any of these functional ingredients. The application of marshmallow root, N-acetyl-D-glucosamine, and maitake mushroom significantly increased the protease activity of S. *thermophilus*, whereas the protease activity of *L*. *bulgaricus* was not affected by any ingredient. Compared to the control, marshmallow root and quercetin samples had higher mean log counts and log counts for *S. thermophilus* on the simulated gastric juice and lysozyme resistance in vitro test, respectively. For *L. bulgaricus*, licorice root, quercetin, marshmallow root, and slippery elm bark samples had higher log counts than the control samples.

## 1. Introduction

Lactic acid bacteria (LAB) play a necessary role in the function and integrity of the gastrointestinal tract by changing the microbial flora and facilitating proper intestinal functioning. Some recent clinical and animal trials have shown that probiotics may have beneficial effects on host health. Advantageous effects of probiotics on improving host health include alleviation of lactose intolerance [1], reducing cholesterol by absorption [2], prevention of chronic gastritis [3], prevention of diarrhea [4], decreasing lumen pH, and inhibiting *Helicobacter* growth and suppressing bacterial growth through direct binding to gram-negative bacteria [5]. Nevertheless, the effective dose is at least 10^6^–10^7^ CFU per g or ml of the product daily [6].

Low gastric pH and the antimicrobial function of pepsin effectively prevent the entry and survival of probiotic bacteria in the intestine. Consequently, survival in such critically acidic conditions is one of the most significant physiological challenges that probiotic cultures must endure by oral administration. Furthermore, probiotics can modify the beta adhesion-2 production in saliva [7]. For this reason, it is essential to evaluate the survivability of probiotics in the saliva, where lysozyme is the main antimicrobial agent. It should also be acknowledged that combining probiotics with other food products may enable such microorganisms to survive during gastric transfer. If the bacteria have survived the gastric barrier, the small intestine environment will be the second significant barrier for probiotic strains to pass through the gastrointestinal tract. Although the small intestinal pH is more desirable for the bacteria’s survival, the presence of pancreatin and bile salts may have disadvantageous effects. *S. thermophilus* and *L. bulgaricus* are used to produce yogurt, which is considered a great vehicle to carry functional ingredients [8,9].

Specific dietary nutrients can help people improve their gut health. Probiotics, vitamins A and D, and amino acids such as glutamine and arginine can reinforce the mucosal barrier and modulate tight junction proteins [10]. Polyphenols significantly impact the inactivation of the NF-κB pathway, a significant regulator of cytokines and interleukins [11]. On the other hand, some edible mushrooms such as *Inonotus obliquus* (chaga mushroom), *Coriolus versicolor* (turkey tail), and *Hericium erinaceus* (Lion’s mane) were found to regulate the gut microbiota by promoting the production of secondary metabolites that have an intestinal epithelial regulation function in cytokine and interleukin production [10]. Medical herbs such as *Camellia sinensis* (tea plant), *Hibiscus sabdariffa* L. (roselle plant), and *Zingiber officinale* (ginger) have significant amounts of bioactive compounds such as organic acids, flavonoids, iridoid glycosides, saponins, chlorogenic acid, secoiridoids, berberine, sesquiterpene, and sesquiterpenoid that can help treat leaky gut-related diseases such as inflammatory bowel disease, obesity, and ulcerative colitis [10]. L-glutamine [12], quercetin [13], slippery elm bark [14], marshmallow root [15], N-acetyl-D-glucosamine (NAG) [16], licorice root [17], maitake mushrooms [18], and zinc orotate [19] have shown potential benefits through gut microbiota and intestinal barrier functions. NAG and licorice have been shown to inhibit pro-inflammatory cytokine production [16,20], while L-glutamine, quercetin, and slippery elm bark can assist in tightening the epithelial junctions in the intestinal walls [12,13,14]. Marshmallow root and zinc orotate show antioxidant and anti-inflammation properties [19,21] and reduce intestinal barrier dysfunctions. On the other hand, maitake mushroom has high amounts of beta-1,6-glucans (immune stimulant) and helps promotes benefits through gut microbiota [18]. These ingredients alongside yogurt can reduce intestinal barrier dysfunctions and beneficially affect the gut microbiota [22]. The effects of these ingredients on the probiotic characteristics of yogurt starter culture bacteria are not well known. Paz et al. (2022) [23] evaluated the probiotic characteristics of *Streptococcus thermophilus* and *Lactobacillus delbrueckii* ssp. *bulgaricus* as affected by carao (*Cassia grandis*), which is a medical plant with great antidiabetic potential [24,25]. The results indicated that carao improved the acid and bile tolerance of yogurt starter culture bacteria [23].

Therefore, this study aimed to evaluate the probiotic properties, tolerance to gastric juices and lysozyme, protease activity, and viability of starter culture bacteria. A series of in vitro analyses were conducted such as acid tolerance, bile tolerance, protease activity, viability, lysozyme resistance, and gastric juice tolerance of *S. thermophilus* and *L. bulgaricus* as influenced by ingredients potentially treating leaky gut.

## 2. Materials and Methods

### 2.1. Experimental Design

Four mild stresses (acid, bile, lysozyme, and gastric juices), protease activity, and viability characteristics were evaluated on starter cultures of *S*. *thermophilus* STI-06 and *L*. *bulgaricus* LB-12 (Chr. Hansen, Milwaukee, WI) separately. Each evaluation had 8 ingredients of testing: (L-glutamine (Y1 = 7 g/L), quercetin (Y2 = 700 mg/L), slippery elm bark (Y3 = 210 mg/L), marshmallow root (Y4 = 1340 mg/L), NAG (Y5 = 210 mg/L), licorice root (Y6 = 210 mg/L), maitake mushrooms (Y7 = 42 mg/L), and zinc orotate (Y8 = 70 mg/L). Each culture containing an ingredient was evaluated against stress (pH 2, oxgall salt (0.3%), lysozyme (100 mg/L), and gastric juice (pepsin (0.32%) and NaCl (0.2%)) and compared to a control (a culture in the absence of the ingredient). The viability test, lysozyme resistance, and gastric juice tolerance were monitored by plate counting. At various times, all test counts were enumerated in MRS agar (*L. bulgaricus*) and M17 agar (*S. thermophilus*). The experiments were repeated 3 times with duplicate readings.

### 2.2. Acid Tolerance Test

The acid tolerance of *S. thermophilus* and *L. bulgaricus* was determined by the method of Perei and Gibson (2002) [26] with some modifications. Starter cultures were inoculated (10% (*v*/*v*)) into acidified MRS broth (Criterion™, Hardy Diagnostics, Santa Maria, CA, USA) previously adjusted to pH 2.0 with 1N HCl. The acidified MRS broth with culture was incubated in a water bath at 37 °C for 15 min. One milliliter samples were taken at various times (0, 30, 60, 90 and 120 min), serially diluted in peptone water, and plated in duplicate into MRS agar (Criterion™, Hardy Diagnostics, Santa Maria, CA, USA) and M17 agar (Criterion™, Hardy Diagnostics, Santa Maria, CA, USA), respectively.

### 2.3. Bile Tolerance Test

Bile tolerance of *S. thermophilus* and *L. bulgaricus* was determined by the method of Perei and Gibson (2002) [26] with some modifications. *S. thermophilus* (10% (*v*/*v*)) was inoculated into M17 broth (Criterion™, Hardy Diagnostics, Santa Maria, CA, USA) supplemented with 0.3% (wt/*v*) oxgall (bovine bile) (US Biological, Swampscott, MA, USA) to evaluate its ability to grow under aerobic conditions at 37 °C. *L. bulgaricus* (10% (*v*/*v*)) was inoculated into MRS broth (Criterion™, Hardy Diagnostics, Santa Maria, CA) supplemented with 0.2% (wt/*v*) sodium thioglycolate (Acros Organics, Fair Lawn, NJ, USA) (an oxygen scavenger to achieve microaerophilic conditions) and 0.3% (wt/*v*) oxgall bile salt to evaluate its ability to grow under anaerobic conditions at 43 °C. Samples of each culture were taken hourly for 8 h and plated.

### 2.4. Protease Activity

The extracellular protease activity of *S. thermophilus* and *L. bulgaricus* was determined by the o-phthaldialdehyde (OPA) spectrophotometric assay according to the method described by Oberg et al. (1991) [27]. *S. thermophilus* and *L. bulgaricus* were inoculated (1% (*v*/*v*)) into sterile skim milk (autoclaved at 121 °C for 15 min), and incubated at 40 °C for 0, 12, and 24 h. After incubation, 2.5 mL from each sample was withdrawn and mixed with 1 mL distilled water. The diluted sample was transferred into test tubes containing 5 mL of 0.75 N trichloroacetic acid (TCA) (Fisher Scientific, Waltham, MA, USA) and immediately vortexed at the same time. After setting at room temperature for 10 min, the acidified samples were filtered through a Whatman Number 2 filter paper (Clifton, NJ, USA). Non-inoculated sterile skim milk was similarly prepared to be used as a blank. The TCA filtrate was analyzed by the o-phthaldialdehyde (OPA) spectrophotometric assay using a UV–Vis spectrophotometer (Nicolet Evolution 100, Thermo Scientific, Madison, WI, USA). The o-phthaldialdehyde final solution was prepared by combining 25 mL of 100 mM sodium borate (Fisher Scientific, Waltham, MA, USA), 2.5 mL of 20% (wt/wt) SDS (Fisher Scientific, Waltham, MA, USA), 40 mg of o-phthaldialdehyde reagent (Alfa Aesar, Ward Hill, MA, USA) dissolved in 1 mL methanol (Sigma, St. Louis, MO, USA), and 100 µL of β-mercaptoethanol, and diluting to a final volume of 50 mL with distilled water. TCA filtrate (150 µL) was mixed with 3 mL of o-phthaldialdehyde final solution in a 3 mL cuvette, and the absorbance at 340 nm was measured.

### 2.5. Microbial Growth

Growth of *S. thermophilus* and *L. bulgaricus* was analyzed by plate counts. *S. thermophilus* was inoculated (10% (*v*/*v*)) into sterile M17 broth (Criterion™, Hardy Diagnostics, Santa Maria, CA, USA), and *L. bulgaricus* was inoculated (10% (*v*/*v*)) into sterile MRS broth (Criterion™, Hardy Diagnostics, Santa Maria, CA, USA). The inoculated M17 and MRS broths were plated as described before (enumeration of *S. thermophilus* and *L. bulgaricus*) and incubated aerobically at 37 °C for 24 h and anaerobically at 43 °C for 72 h, respectively. The plating took place hourly.

### 2.6. Tolerance to Simulated Gastric Juice

Tolerance of *S. thermophilus* and *L. bulgaricus* with the functional ingredients to simulated gastric juice (SGJ) was conducted according to García-Ruiz et al. (2014) [28] and Zhang et al. (2019) [29] with slight modifications. SGJ was formulated with autoclaved H_2_O, filter-sterilized pepsin (Sigma-Aldrich, St. Louis, MO, USA) (0.32%), and autoclaved NaCl solution (0.2%), while NaOH and HCl were utilized to adjust the pH. The SGJ was adjusted to five pH values (pH 7.0, 5.0, 4.0, 3.0, and 2.0) with 1 M HCl and 1 M NaOH. Starter cultures of *S. thermophilus* and *L. bulgaricus* were inoculated (5% (*v*/*v*)), individually, into SGJ and incubated under aerobic conditions at 37 °C (*S. thermophilus*) and anaerobically at 43 °C (*L. bulgaricus*) for 30 min. The counts of viable bacteria were enumerated by plate counting at 0 and 30 min of incubation.

### 2.7. Lysozyme Tolerance Test

Resistance of *S. thermophilus* and *L. bulgaricus* to lysozyme was determined by the method described by Zago et al. (2011) [30] with modification. Cultures were evaluated for their capacity to survive in a filter-sterilized electrolyte solution (0.22 g/L CaCl_2_, 6.2 g/L NaCl, 2.2 g/L KCl, 1.2 g/L NaHCO_3_) in the presence of 100 mg/L of lysozyme (Sigma Aldrich, St. Louis, MO, USA) (Vizoso-Pinto et al. 2006) [31]. Starter cultures were inoculated (10% (*v*/*v*)) into the electrolyte solution and incubated under aerobic conditions at 37 °C (*S. thermophilus*) and anaerobically at 43 °C (*L. bulgaricus*). The counts of viable bacteria were enumerated by plate counting at 0, 30, and 120 min of incubation.

### 2.8. Enumeration of S. thermophilus

*S. thermophilus* agar was prepared by weighing 10 g of sucrose (Amresco, Solon, OH, USA), 2 g of K2HPO4 (Fisher Scientific, Fair Lawn, NJ, USA), 5 g of Bacto yeast extract and 10 g of Bacto Tryptone (Becton, Dickinson and Co., Sparks, MD, USA) per L of distilled water. The pH was reduced to pH 6.8 before adding 12 g of agar (Fisher Scientific, Fair Lawn, NJ, USA) and 6 mL of 0.5% bromocresol purple (Fisher Scientific, Fair Lawn, NJ, USA). Samples were serially diluted with 99 mL of sterilized MgCl_2_ and KOH, pour plated with autoclaved media, and aerobically incubated at 37 °C for 24 h. To enumerate the colonies, a Quebec Darkfield Colony Counter (Leica Inc., Buffalo, NY, USA) was used.

### 2.9. Enumeration of L. bulgaricus

The MRS agar was prepared by weighing 55 g of Lactobacilli MRS broth powder (Becton, Dickinson and Co., Sparks, MD, USA) and 15 g of agar (Fisher Scientific, Fair Lawn, NJ, USA) per liter of media. The pH was adjusted to 5.2 using 1 N HCl. The samples were serially diluted with 99 mL of sterilized MgCl_2_ and KOH, pour plated, and incubated anaerobically in a BBL GasPak (BBL, Becton, Dickinson and Co., Cockeysville, MD, USA) at 43 °C for 72 h. Colonies were enumerated with a Quebec Darkfield Colony Counter (Leica Inc., Buffalo, NY, USA).

## 3. Statistical Analysis

Data were analyzed using the general linear model (PROC GLM) of the Statistical Analysis Systems (SAS). A two-factor factorial experiment in a randomized block design was applied in the experimental design. Differences of least square means were used for main effects (ingredient and time) and interaction effect (ingredient × time) at α = 0.05. Tukey’s test examined the statistical differences (*p* < 0.05) among the main and interaction effects.

## 4. Results and Discussion

### 4.1. Acid Tolerance

The acid tolerance of *S. thermophilus* as affected by the incorporation of the ingredients over 120 min of incubation is illustrated in Figure 1. The ingredient effect, time effect, and ingredient × time interaction effect were significant (*p* < 0.05) (Table 1). With acid exposure, the log count significantly declined from 0 to 30 min, but remained stable from 30 to 120 min. The log counts for the marshmallow and licorice root samples were higher than the control from 30 to 120 min, whereas the L-glutamine and slippery elm bark samples had lower counts than the control (Table 2). The mean log difference varied from 3.82 (N-acetyl-D-glucosamine) to 4.75 (Quercetin) among control samples and ingredient treatments (Table 3). The log difference results from 0 h to 2 h showed that the control and ingredients samples had less than 1 log of difference. When using log difference from 0 h to 2 h, there were no differences (*p* > 0.05) among treatments. The least square difference (Table 4) showed slightly higher counts (*p* < 0.05) for marshmallow and licorice root samples when compared with all treatments.

Figure 2 presents the acid tolerance of *L. bulgaricus* as influenced by adding the ingredients over 120 min of incubation. The ingredient effect and ingredient × time interaction effect were not significant (*p* > 0.05), whereas the time effect was significant (Table 1 and Table 2). The log count for all ingredients significantly decreased with acid exposure from 0 to 120 min and followed the same decreasing pattern as the control samples.

Many reports have revealed varying effects concerning the effect of polyphenols and herbs on the growth of probiotics. Licorice root may contain mucilaginous materials that can coat probiotics and reduce gastric acidity, improve gastric emptying, and promote gastric healing [32]. Similarly, the main active compounds from *Althaeae* sp. root are polysaccharides that contain high quantities of starch (37%), mucilages (15–35%), and pectins (10–12%) [33]. These components can be coating agents that can protect the probiotics. On the other hand, anti-microbial activity of medicinal plants may be due to the presence of specific polyphenols, isoprenoids, alkaloids, steroids, and saponins [34]. Other studies have shown that slippery elm has a bactericidal agent against *Streptococcus pyogenes* strains. In *S. thermophilus* CHCC 3534, the strain showed tolerance to 0.8% gallic acid and 0.3% catechin at low pH (2.0). The compounds flavan-3-ols and anthocyanins in wine extract did not provide a significant impact on species of *Lactobacillus*, *Enterococcus*, *Bacteroides*, and *Bifidobacterium* genera [35]. Chan et al., (2018) [36] noted no inhibitory results of phenolic-rich extracts of spices and medicinal plants under 313–2500 μg/mL against *Lactobacillus* species. In addition, *Lactobacillus acidophilus* CECT 903 was not inhibited by quercetin, tannic acid, gallic acid, caffeic acid, catechin, and epicatechin under 5000 μg/disk [37]. Nevertheless, high amounts of gallate-derived compounds [e.g., (−)-epicatechin-3-O-gallate] have shown inhibitory effects on the growth of *L. bulgaricus* [38]. The inhibitory effect of L-glutamine on *S. thermophilus* could be related to its ability at low pH to cause alterations in the structure of the cytoplasmic membrane, with changes in polarization and permeability at high concentrations.

### 4.2. Bile Tolerance

The bile tolerance over 8 h of incubation of *S. thermophilus* and L. *bulgaricus* as affected by the incorporation of the ingredients is illustrated in Figure 3 and Figure 4, respectively. For both *S. thermophilus* and *L. bulgaricus*, the ingredient effect, time effect, and ingredient × time interaction effect were significant (*p* < 0.05) (Table 1). L-glutamine had significantly (*p* < 0.05) lower log counts compared to the control for *S. thermophilus* (at 0, 4, and 8 h) and *L. bulgaricus* (at 4 h) (Table 2).

Bile salts usually impact the survival of bacteria, and the ability of probiotic strains to hydrolyze bile salts has often been included among the criteria for probiotic strain selection [39]. Bile salts can cross the cell membrane, damage proteins and DNA, and result in the leakage of intracellular material [40]. Vargas et al. (2015) [41] noted bile tolerance for *L. bulgaricus* and *S. thermophilus* at 8 h of incubation in MRS broth and M17 broth containing 0.3% oxgall, respectively. They found that whey protein decreased cell reduction in broth with 0.3% oxgall compared with the control. According to Ziegler et al. (1992) [42], glutamine can enhance the activity of digestive enzymes. A reduction in counts is expected due to the exposure of phospholipid cell walls of bacteria to bile salts, which dissolve the phospholipid structure [39]. Herbs may restrict the damage to bacterial cell proteins or promote protein repair. They may serve as a barrier or a partial barrier as mucilaginous materials can coat the bile and lipid membrane of bacterial cells [31]. The inhibitory effects that can be enhanced by polyphenolic compounds such as quercetin on probiotics can be related to their ability to cause alterations in the structure of cytoplasmatic membranes with changes in polarization and permeability [37].

### 4.3. Microbial Growth

The growth of *S. thermophilus* STI-06 over 16 h of incubation as affected by the incorporation of functional ingredients is depicted in Figure 5. The ingredient effect, time effect, and ingredient × time interaction effect were significant (*p* < 0.05) (Table 1). For control samples, the log count increased from 8.68 to 9.10 during the first 2 h and minimally decreased from 9.10 to 8.65 between 2 and 16 h. The slippery elm bark, marshmallow root, N-acetyl-D-glucosamine, licorice root, maitake mushrooms, zinc orotate, and quercetin followed an identical trend (Table 4) as the control samples by growth from 0 to 2, a relatively rapid decrease from 2 to 4 h, followed by a slower decrease from 4 to 16 h. L-glutamine had a slight decline from 0 to 2 h, a more rapid decline from 2 to 4 h, and only slight changes between 4 and 16 h.

The growth of *L. bulgaricus* LB-12 during 16 h of incubation as affected by the incorporation of functional ingredients is illustrated in Figure 6. The ingredient effect, time effect, and ingredient × time interaction effect were significant (*p* < 0.05) (Table 1). The log count of control samples increased from 7.85 to 8.37 during the first 2 h and decreased from 8.37 to 7.61 between 2 and 16 h. The slippery elm bark, marshmallow root, N-acetyl-D-glucosamine, licorice root, maitake mushrooms, quercetin, and zinc orotate, followed an identical growth trend as the control samples (Table 4). L-glutamine had a slight decline from 0 to 2 h, a more rapid decline from 2 to 4 h, and only slight changes between 4 and 16 h. On the other hand, quercetin showed higher log counts when compared to control samples.

There is little information regarding the effects of these ingredients on probiotic characteristics. Commonly, strains of *S. thermophilus* need essential amino acids and peptides to grow [43]. *S. thermophilus* is less susceptible to a shortage of amino acids compared to *Lactobacilli* strains [43]. Herbs are known as preservatives and medicine and it is due to substances such as flavonoids, anthocyanins, alkaloids, glycosides, saponins, coumarins, tannins, vitamins, phenolic acids, and many more. Many studies have revealed inconsistent results regarding the impact of polyphenols on the growth of lactic acid bacteria. The phenolic extracts from blueberry, lingonberry, blackcurrant, raspberry, cloudberry, cranberry, and strawberry have shown inhibition against *Lactobacillus rhamnosus* and *L. rhamnosus*, while it did not show an effect in *Lactobacillus plantarum* [44]. In addition, flavan-3-ol enriched grape seed extract inhibits the growth of *S. thermophilus*, *Bifidobacterium lactis* BB12, *Lactobacillus fermentum*, *L. acidophilus*, and *L. vaginalis*, whereas it stimulates the growth of some *Lactobacillus plantarum*, *L. casei*, and *L. bulgaricus* strains. It has no impact on *Bifidobacterium breve* 26M2 and *B. bifidum* HDD541 growth [38]. In our study, none of the herbs, polyphenols or amino acids affected the growth of *L. bulgaricus* LB-12 and *S. thermophilus* STI-06 during 16 h of storage, meaning that these functional ingredients could be recommended to be used at given concentrations with these lactic acid bacteria in the yogurt fermentation process.

### 4.4. Protease Activity

The least square means for protease activity (absorbance) of *S. thermophilus* STI-06 over an incubation period of 24 h as affected by the incorporation of the functional ingredients are shown in Table 5. The ingredient effect, time effect, and ingredient × time interaction effect were significant (*p* < 0.05). The addition of marshmallow root, N-acetyl-D-glucosamine, and maitake mushroom showed significantly (*p* < 0.05) higher protease activity than the control at 12 and 24 h of incubation. At 24 h of incubation, all treatments had significantly (*p* < 0.05) higher protease activity than at 0 and 12 h. At 12 and 24 h, marshmallow root (0.247 and 0.320) had the highest protease activity. This result is not surprising since marshmallow root had the highest value for *S. thermophilus* log counts (Figure 5) and this high cell density could lead to high protease activity [45]. Additionally, marshmallow root (*Althaea officinalis*) has high amounts of polysaccharides [32], which could be used by *S. thermophilus* for its growth and its increased protease activity. *S. thermophilus* would likely convert N-acetyl-D-glucosamine to amino acids, since others studies have shown that yeasts and *S. stipitis* NBRC10063 can ferment N-acetyl-D-glucosamine to amino acids [46].

The least square means for protease activity (absorbance) of *L. bulgaricus* LB-12 over an incubation period of 24 h as impacted by the incorporation of the functional ingredients are illustrated in Table 6. Time effect was significant (*p* < 0.05) whereas the ingredient effect and ingredient × time interaction effect was not significant (*p* > 0.05). Protease activity significantly increased from 0 to 24 h.

*L. bulgaricus* and *S. thermophilus* grow synergistically. *S. thermophilus* produces folic acid that facilitates the growth of *L. bulgaricus*, and *L. bulgaricus* metabolizes proteins to free amino acids that enhance the growth of *S. thermophilus* [47]. Not surprisingly, higher protease activity was observed in *L. bulgaricus* than in *S. thermophilus*. Other studies have stated similar results when comparing the protease activity of *L. bulgaricus* to *S. thermophilus* [43].

### 4.5. Tolerance to Simulated Gastric Juice (SGJ)

To accurately simulate the resistance to upper gastrointestinal transit, the cultures were exposed to simulated gastric juices with the functional ingredients. The counts of *S. thermophilus* as impacted by the addition of functional ingredients into simulated gastric juice over 30 min of incubation at pH 2, 3, 4, 5, and 7 are illustrated in Figure 7. The ingredient effect, pH effect, and ingredient × pH interaction effect were significant (*p* < 0.05) (Table 1). There was no significant difference (*p* > 0.05) when pH was 5.0 versus 7.0 for all treatments, including the control samples, except for quercetin and marshmallow root samples, which had the higher log mean counts. Over 30 min of incubation, the log mean counts of all treatments decreased slightly when the pH was 4.0. When the simulated gastric juice was pH 2.0 and 3.0, the L-glutamine log counts were significantly (*p* < 0.05) lowered compared to the other functional ingredients, including the control samples (Table 4).

In Figure 8, the SGJ tolerance of *L. bulgaricus* as influenced by the addition of the ingredients over 30 min of incubation at different pH values is presented. The ingredient effect and ingredient × pH interaction effect were not significant (*p* > 0.05), whereas the pH effect was significant (*p* < 0.05) (Table 1). The addition of the ingredients did not decrease the log counts when compared to the control. The log counts for all ingredients followed a similar pattern (Table 4) where the log means count decreased as pH decreased below pH 5. Zhang et al. (2020) [29] reported a similar trend, as probiotic bacteria are susceptible to acidic stress.

The digestive tract is a harsh environment where pepsin and salts are the main components of the low pH gastric juices [48]. In other studies, *S. thermophilus* had shown a greater tolerance than *L. bulgaricus* to simulated gastric juice [49]. The results for *S. thermophilus* indicated that quercetin and marshmallow root samples have a greater capacity to survive in gastric juice while L-glutamine samples have a lower possibility to survive in gastric juice. *Althaea officinalis* is a polysaccharide that has the capacity to effectively bind to mucous tissue and has great coating ability [50]. Possibly, the *Althaea officinalis* polysaccharides could protect *S. thermophilus* by coating the viable bacteria in the gastric juices. The possible protective effects of quercetin could be attributed to its antioxidant properties, which could protect probiotic cells from the damage caused by exposure to the harsh conditions found in the gastrointestinal tract [51]. On the other hand, the results obtained for *L. bulgaricus* indicated that the ingredients exerted no negative impact on the survival when exposed to simulated in vitro digestion. The fact that the ingredients did not exert any negative impact on the survival of *L. bulgaricus* is an interesting finding because only the bioactive components that resist the stomach and small intestine conditions can reach the large intestine and exert their beneficial effects on the host.

### 4.6. Resistance to Lysozyme

The lysozyme resistance of *S. thermophilus* as affected by ingredients that help treat leaky gut is presented in Figure 9. The ingredient effect, time effect, and ingredient × time interaction effect were significant (*p* < 0.05) (Table 1). In 90 and 120 min, the log counts of control samples significantly (*p* < 0.05) declined to around 4.56 log CFU/mL, whereas the log counts of quercetin and marshmallow root decreased to about 5.65 log CFU/mL. In addition, slippery elm bark, L-glutamine, N-acetyl-D-glucosamine, licorice root, maitake mushrooms, and zinc orotate also showed similar lysozyme resistance compared to the control (Table 4). The mean log count difference varied from 2.93 (L glutamine) to 3.77 (slippery elm bark) among control samples and ingredient treatments (Table 3). Results indicated that the treatments had less than 1 log of difference. There were no differences (*p* > 0.05) when using log difference from 0 h to 2 h among treatments. The least square difference (Table 4) showed slightly higher (*p* < 0.05) counts for quercetin and marshmallow root samples when compared to all treatments.

Lysozyme resistance of *L. bulgaricus* as influenced by ingredients that help treat leaky gut is shown in Figure 10. The ingredient effect, time effect, and ingredient × time interaction effect were significant (*p* < 0.05). In 30, 60, 90, and 120 min, quercetin, licorice root, slippery elm bark, and marshmallow root samples had significantly (*p* < 0.05) higher log counts than the control samples (Table 6).

Higher counts and survival rates for indicated ingredients for both bacteria showed they could have increased lysozyme tolerance. For prebiotic selection, it is necessary to consider the resistance to the severe conditions of the gastrointestinal tract. The human mouth is the first obstacle that probiotics encounter, and there is a high amount of lysozyme in the oral cavity [28]. In this study, quercetin and marshmallow root samples similarly showed the highest counts in the tolerance to simulated gastric juice experiment. Once again, the protective quercetin and marshmallow root were established in the lysozyme resistance experiment for these bacteria. Since there is a lack of studies concerning inhibiting or growing selected microorganisms for the selected ingredients, all ingredients at given concentrations showed no adverse effect on lysozyme tolerance, meaning that these ingredients do not interfere with the survival of *S. thermophilus* and *L. bulgaricus* in the oral cavity.

## 5. Conclusions

The effect of ingredients that help treat leaky gut on the probiotic properties of yogurt starter cultures was investigated. Overall, the results obtained from this study indicated that the incorporation of these ingredients at given concentrations did not adversely affect the probiotic properties of yogurt starter bacteria. The application of marshmallow root, licorice root, and slippery elm bark resulted in significantly higher acid and bile tolerance for *S. thermophilus* STI-06, whereas L-glutamine, quercetin, slippery elm bark, marshmallow root, N-acetyl-D-glucosamine, licorice root, maitake mushrooms, and zinc orotate followed a similar trend as the control broths for acid and bile tolerance for *L. bulgaricus* LB-12. The growth of *S. thermophilus* and *L. bulgaricus* was not affected by any of these functional ingredients. The application of marshmallow root, licorice root, and slippery elm bark significantly increased protease activity of *S. thermophilus* over 24 h of incubation, whereas the protease activity of *L. bulgaricus* was not affected by any ingredient. Tolerance of *L. bulgaricus* to simulated gastric juice was not affected by any ingredients. In *S. thermophilus*, quercetin and marshmallow root samples had higher mean log counts and log counts on the simulated gastric juice and lysozyme resistance in the in vitro test than the control. Quercetin, licorice root, slippery elm bark, and marshmallow root samples had higher log counts for *L. bulgaricus* than the control samples. In conclusion, these functional ingredients at recommended concentrations can be applied in yogurt production.

## Figures and Tables

**Figure 1 microorganisms-11-00893-f001:**
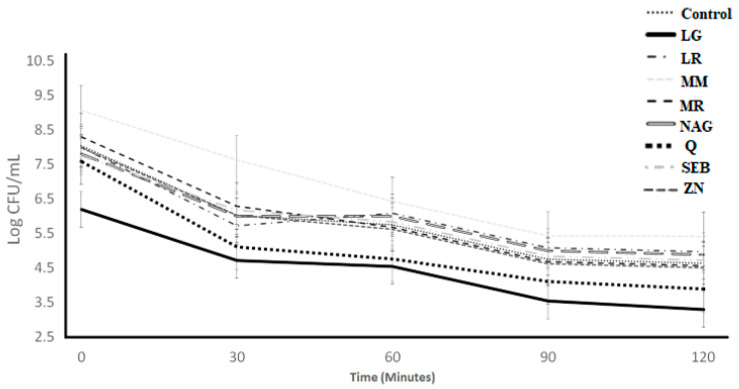
Log counts of *Streptococcus thermophilus* showing its acid tolerance as influenced by different ingredients over an incubation period of 120 min. C = control, LG = L-glutamine (7 g/L), Q = quercetin (700 mg/L), SEB = slippery elm bark (210 mg/L), MR = marshmallow root (1340 mg/L), NAG = N-acetyl-D-glucosamine (210 mg/L), LR = licorice root (210 mg/L), MM = maitake mushrooms (42 mg/L), and ZN = zinc orotate (70 mg/L). Error bars represent SE.

**Figure 2 microorganisms-11-00893-f002:**
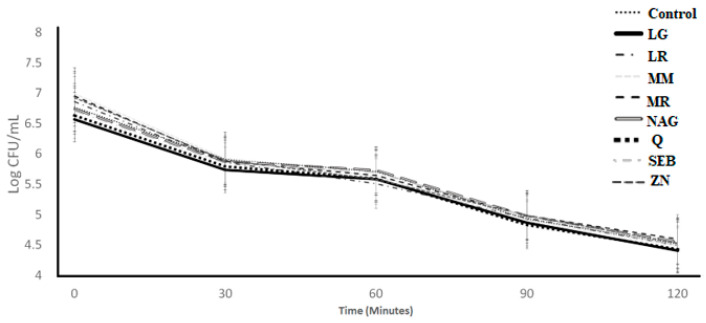
Log counts of *Lactobacillus bulgaricus* showing its acid tolerance as influenced by different ingredients over an incubation period of 120 min. C = control, LG = L-glutamine (7 g/L), Q = quercetin (700 mg/L), SEB = slippery elm bark (210 mg/L), MR = marshmallow root (1340 mg/L), NAG = N-acetyl-D-glucosamine (210 mg/L), LR = licorice root (210 mg/L), MM = maitake mushrooms (42 mg/L), and ZN = zinc orotate (70 mg/L). Error bars represent SE.

**Figure 3 microorganisms-11-00893-f003:**
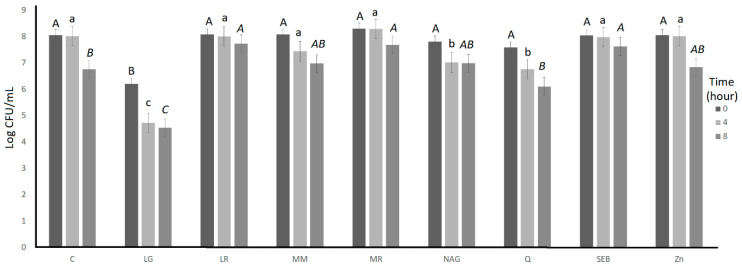
Log counts of *Streptococcus thermophilus* showing its bile tolerance as influenced by treatments over 8 h. C = control, LG = L-glutamine (7 g/L), Q = quercetin (700 mg/L), SEB = slippery elm bark (210 mg/L), MR = marshmallow root (1340 mg/L), NAG = N-acetyl-D-glucosamine (210 mg/L), LR = licorice root (210 mg/L), MM = maitake mushrooms (42 mg/L), and ZN = zinc orotate (70 mg/L). Average of three replicates. ^A,B^ Values with different letters at 0 h are significantly different (*p* < 0.05). ^a,b,c^ Values with different letters at 4 h are significantly different (*p* < 0.05). *^A,B,C^* Values with different letters at 8 h are significantly different (*p* < 0.05). Error bars represent SE.

**Figure 4 microorganisms-11-00893-f004:**
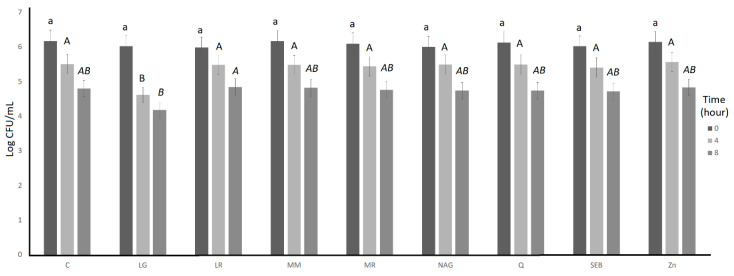
Log counts of *Lactobacillus bulgaricus* showing its bile tolerance as influenced by ingredients over 8 h. C = control, LG = L-glutamine (7 g/L), Q = quercetin (700 mg/L), SEB = slippery elm bark (210 mg/L), MR = marshmallow root (1340 mg/L), NAG = N-acetyl-D-glucosamine (210 mg/L), LR = licorice root (210 mg/L), MM = maitake mushrooms (42 mg/L), and ZN = zinc orotate (70 mg/L). Average of three replicates. ^a^: Values with different letters at 4 h are significantly different (*p* < 0.05). ^A^ There were no significant (*p* > 0.05) differences at 0 h. ^A,B,^ Values with different letters at 4 h are significantly different (*p* < 0.05). *^A,B^* Values with different letters at 8 h are significantly different (*p* < 0.05). Error bars represent SE.

**Figure 5 microorganisms-11-00893-f005:**
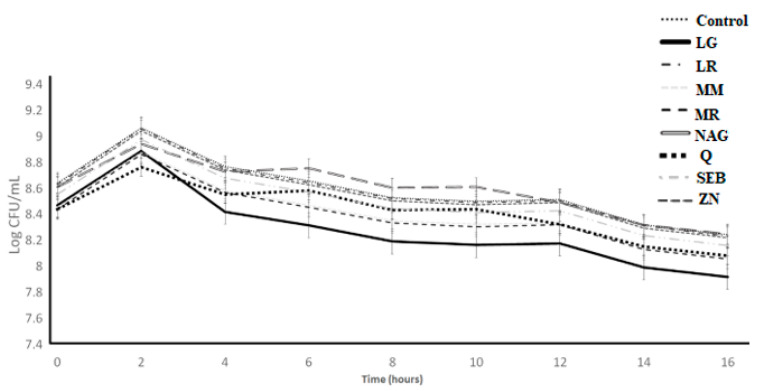
Log counts of *Streptococcus thermophilus* showing its growth as influenced by ingredients over the incubation period of 16 h. C = control, LG = L-glutamine (7 g/L), Q = quercetin (700 mg/L), SEB = slippery elm bark (210 mg/L), MR = marshmallow root (1340 mg/L), NAG = N-acetyl-D-glucosamine (210 mg/L), LR = licorice root (210 mg/L), MM = maitake mushrooms (42 mg/L), and ZN = zinc orotate (70 mg/L). Error bars represent SE.

**Figure 6 microorganisms-11-00893-f006:**
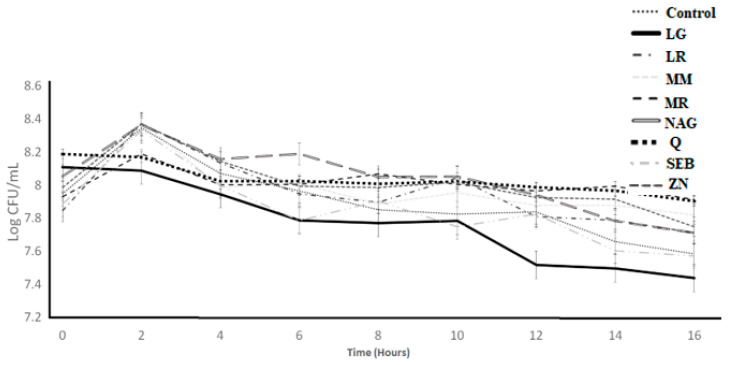
Log counts of *Lactobacillus bulgaricus* showing its growth as influenced by ingredients over the incubation period of 16 h. C = control, LG = L-glutamine (7 g/L), Q = quercetin (700 mg/L), SEB = slippery elm bark (210 mg/L), MR = marshmallow root (1340 mg/L), NAG = N-acetyl-D-glucosamine (210 mg/L), LR = licorice root (210 mg/L), MM = maitake mushrooms (42 mg/L), and ZN = zinc orotate (70 mg/L). Error bars represent SE.

**Figure 7 microorganisms-11-00893-f007:**
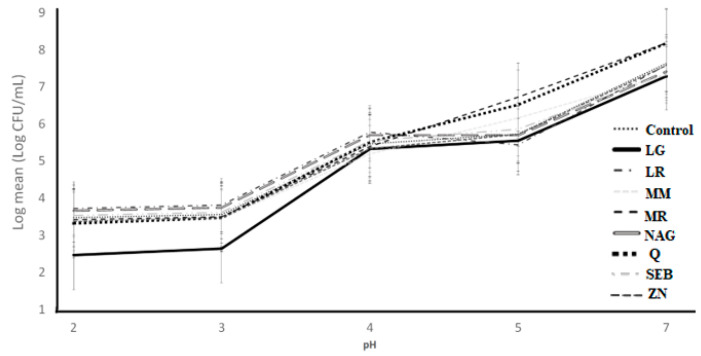
Log counts of *Streptococcus thermophilus* showing its resistance to simulated gastric juice as influenced by ingredients over different pH conditions. C = control, LG = L-glutamine (7 g/L), Q = quercetin (700 mg/L), SEB = slippery elm bark (210 mg/L), MR = marshmallow root (1340 mg/L), NAG = N-acetyl-D-glucosamine (210 mg/L), LR = licorice root (210 mg/L), MM = maitake mushrooms (42 mg/L), and ZN = zinc orotate (70 mg/L). Error bars represent SE.

**Figure 8 microorganisms-11-00893-f008:**
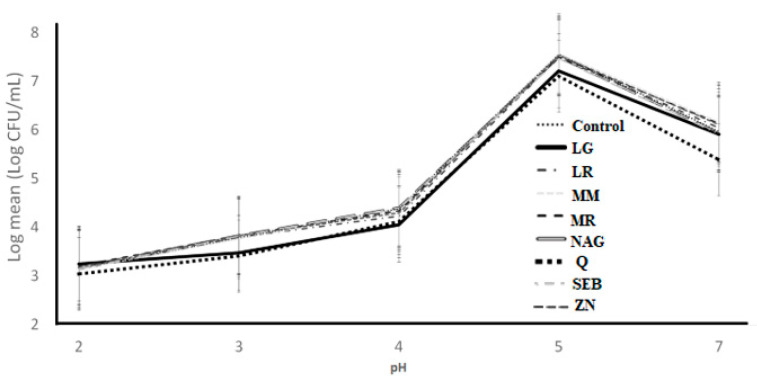
Log counts of *Lactobacillus bulgaricus* showing its resistance to simulated gastric juice as influenced by ingredients over different pH conditions. C = control, LG = L-glutamine (7 g/L), Q = quercetin (700 mg/L), SEB = slippery elm bark (210 mg/L), MR = marshmallow root (1340 mg/L), NAG = N-acetyl-D-glucosamine (210 mg/L), LR = licorice root (210 mg/L), MM = maitake mushrooms (42 mg/L), and ZN = zinc orotate (70 mg/L). Error bars represent SE.

**Figure 9 microorganisms-11-00893-f009:**
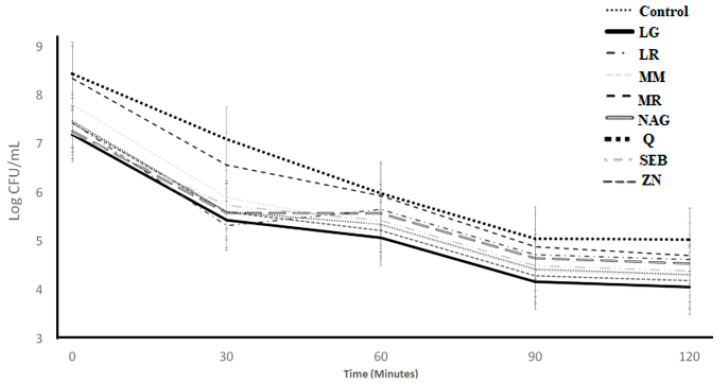
Log counts of *Streptococcus thermophilus* showing its resistance to lysozyme as influenced by ingredients during an incubation time of 120 min. C = control, LG = L-glutamine (7 g/L), Q = quercetin (700 mg/L), SEB = slippery elm bark (210 mg/L), MR = marshmallow root (1340 mg/L), NAG = N-acetyl-D-glucosamine (210 mg/L), LR = licorice root (210 mg/L), MM = maitake mushrooms (42 mg/L), and ZN = zinc orotate (70 mg/L). Error bars represent SE.

**Figure 10 microorganisms-11-00893-f010:**
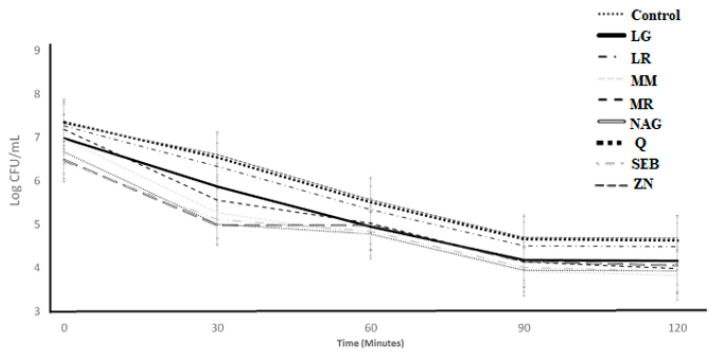
Log counts of *Lactobacillus bulgaricus* showing its resistance to lysozyme as influenced by ingredients during an incubation time of 120 min. C = control, LG = L-glutamine (7 g/L), Q = quercetin (700 mg/L), SEB = slippery elm bark (210 mg/L), MR = marshmallow root (1340 mg/L), NAG = N-acetyl-D-glucosamine (210 mg/L), LR = licorice root (210 mg/L), MM = maitake mushrooms (42 mg/L), and ZN = zinc orotate (70 mg/L). Error bars represent SE.

**Table 1 microorganisms-11-00893-t001:** The *p*-value > *F*-value of ingredient, time or pH, and their interaction for bacterial viability, bile tolerance, acid tolerance, resistance to gastric juices, protease activity, and lysozyme resistance *S. thermophilus* and *L. bulgaricus*.

Effect	*S. thermophilus* STI-06	*L. bulgaricus* LB-12
**Viability**		
Ingredient	<0.0001	<0.0001
Time (Hours)	<0.0001	<0.0001
Ingredient × time	<0.0001	<0.0001
**Bile tolerance**		
Ingredient	<0.0001	<0.0001
Time (Hours)	<0.0001	<0.0001
Ingredient × time	<0.0001	<0.0001
**Acid tolerance**		
Ingredient	<0.0001	0.0576
Time (Minutes)	<0.0001	<0.0001
Ingredient × time	<0.0001	0.0775
**Resistance to gastric juices**		
Ingredient	<0.0001	0.0765
pH	<0.0001	<0.0001
Ingredient × pH	<0.0001	0.1450
**Protease activity**		
Ingredient	<0.0001	0.0579
Time (Hours)	<0.0001	<0.0001
Ingredient × time	<0.0001	0.4460
**Lysozyme resistance**		
Ingredient	<0.0001	<0.0001
Time (Minutes)	<0.0001	<0.0001
Ingredient × time	<0.0001	<0.0001

**Table 2 microorganisms-11-00893-t002:** Least squares means for acid and bile tolerance of *L. bulgaricus* and *S. thermophilus* as influenced by ingredients.

Ingredient	*S. thermophilus*	*L. bulgaricus*
**Acid tolerance**		
C	5.848 ^c^	NS
LG	4.460 ^d^	NS
Q	5.790 ^c^	NS
SEB	5.054 ^c^	NS
MR	5.908 ^bc^	NS
NAG	5.944 ^bc^	NS
LR	5.977 ^ab^	NS
MM	6.003 ^a^	NS
ZN	5.866 ^c^	NS
**Bile tolerance**		
C	7.656 ^a^	5.543 ^a^
LG	5.132 ^b^	4.960 ^b^
Q	7.087 ^a^	5.124 ^a^
SEB	7.654 ^a^	5.254 ^a^
MR	7.754 ^a^	5.548 ^a^
NAG	7.287 ^a^	5.439 ^a^
LR	7.940 ^a^	5.487 ^a^
MM	7.476 ^a^	5.487 ^a^
ZN	7.245 ^a^	5.557 ^a^

^a–c^: Means within a column not containing a common superscript differ (*p* < 0.05). C = Control, LG = L-glutamine (7 g/L), Q = quercetin (700 mg/L), SEB = slippery elm bark (210 mg/L), MR = marshmallow root (1340 mg/L), NAG = N-acetyl-D-glucosamine (210 mg/L), LR = licorice root (210 mg/L), MM = maitake mushrooms (42 mg/L), and ZN = zinc orotate (70 mg/L). NS= not significant.

**Table 3 microorganisms-11-00893-t003:** Mean log difference (log cfu/mL) in the acid tolerance and resistance to lysozyme counts of *Streptococcus salivarius* ssp. *thermophilus* ST-M5.

Ingredient	*S. thermophilus*
**Acid Tolerance**	
C	3.72
LG	2.93
Q	3.53
SEB	3.77
MR	2.99
NAG	2.99
LR	3.02
MM	3.70
ZN	3.07
**Resistance to Lysozyme**	
C	4.01
LG	4.18
Q	4.77
SEB	4.23
MR	4.49
NAG	3.82
LR	3.87
MM	3.92
ZN	3.89

Mean log difference for acid tolerance and lysozyme resistance = (viable log cfu/mL at 0 h)—(viable log cfu/mL at 2 h). C = control, LG = L-glutamine (7 g/L), Q = quercetin (700 mg/L), SEB = slippery elm bark (210 mg/L), MR = marshmallow root (1340 mg/L), NAG = N-acetyl-D-glucosamine (210 mg/L), LR = licorice root (210 mg/L), MM = maitake mushrooms (42 mg/L), and ZN = zinc orotate (70 mg/L).

**Table 4 microorganisms-11-00893-t004:** Least squares means for viability and gastric juices tolerance of *L. bulgaricus* and *S. thermophilus* as influenced by ingredients.

Ingredient	*S. thermophilus*	*L. bulgaricus*
**Viability**		
C	8.772 ^a^	7.965 ^a^
LG	8.578 ^b^	7.645 ^b^
Q	8.832 ^a^	7.95 ^a^
SEB	8.876 ^a^	7.834 ^a^
MR	8.765 ^a^	8.023 ^a^
NAG	8.786 ^a^	7.934 ^a^
LR	8.785 ^a^	7.911 ^a^
MM	8.865 ^a^	7.845 ^a^
ZN	8.874 ^a^	7.928 ^a^
**Resistance to gastric juices**		
C	5.176 ^a^	NS
LG	4.654 ^b^	NS
Q	5.434 ^a^	NS
SEB	5.156 ^a^	NS
MR	5.467 ^a^	NS
NAG	5.215 ^a^	NS
LR	5.237 ^a^	NS
MM	5.125 ^a^	NS
ZN	5.190 ^a^	NS
**Resistance to lysozyme**		
C	5.006 ^b^	4.861 ^b^
LG	4.967 ^b^	4.811 ^b^
Q	5.562 ^a^	5.785 ^a^
SEB	5.102 ^b^	5.766 ^a^
MR	5.498 ^a^	5.531 ^a^
NAG	5.054 ^b^	4.954 ^b^
LR	5.106 ^b^	5.544 ^a^
MM	5.142 ^b^	5.046 ^b^
ZN	5.121 ^b^	4.883 ^b^

^a,b^: Means within a column not containing a common superscript differ (*p* < 0.05). C = control, LG = L-glutamine (7 g/L), Q = quercetin (700 mg/L), SEB = slippery elm bark (210 mg/L), MR = marshmallow root (1340 mg/L), NAG = N-acetyl-D-glucosamine (210 mg/L), LR = licorice root (210 mg/L), MM = maitake mushrooms (42 mg/L), and ZN = zinc orotate (70 mg/L). NS= not significant.

**Table 5 microorganisms-11-00893-t005:** Least square means for protease activity (absorbance) of *Streptococcus thermophilus* STI-06 over an incubation period of 24 h as influenced by added ingredients.

Sample	0 h	12 h	24 h
C	0.148 ± 0.005 ^a^	0.155 ± 0.005 ^d^	0.202 ± 0.011 ^c^
LG	0.151 ± 0.013 ^a^	0.170 ± 0.013 ^c^	0.221 ± 0.017 ^c^
Q	0.155 ± 0.019 ^a^	0.205 ± 0.019 ^b^	0.217 ± 0.024 ^c^
SEB	0.154 ± 0.021 ^a^	0.186 ± 0.021 ^c^	0.232 ± 0.028 ^bc^
MR	0.150 ± 0.023 ^a^	0.247 ± 0.023 ^a^	0.320 ± 0.027 ^a^
NAG	0.157 ± 0.012 ^a^	0.207 ± 0.012 ^b^	0.271 ± 0.015 ^b^
LR	0.152 ± 0.010 ^a^	0.175 ± 0.010 ^c^	0.227 ± 0.022 ^c^
MM	0.155 ± 0.017 ^a^	0.192 ± 0.017 ^bc^	0.249 ± 0.013 ^b^
ZN	0.150 ± 0.007 ^a^	0.173 ± 0.007 ^c^	0.225 ± 0.022 ^c^

^a–d^ Column means not containing a common letter are significantly (*p* < 0.05) different. C = control, LG = L-glutamine (7 g/L), Q = quercetin (700 mg/L), SEB = slippery elm bark (210 mg/L), MR = marshmallow root (1340 mg/L), NAG = N-acetyl-D-glucosamine (210 mg/L), LR = licorice root (210 mg/L), MM = maitake mushrooms (42 mg/L), and ZN = zinc orotate (70 mg/L).

**Table 6 microorganisms-11-00893-t006:** Least square means for protease activity (absorbance) of *Lactobacillus bulgaricus* LB-12 over an incubation period of 24 h as influenced by added ingredients.

Sample	0 h	12 h	24 h
C	0.160 ± 0.009	0.313 ± 0.006	0.414 ± 0.017
LG	0.163 ± 0.007	0.320 ± 0.013	0.433 ± 0.021
Q	0.162 ± 0.005	0.335 ± 0.020	0.447 ± 0.016
SEB	0.159 ± 0.005	0.327 ± 0.015	0.441 ± 0.012
MR	0.160 ± 0.006	0.321 ± 0.017	0.427 ± 0.023
NAG	0.164 ± 0.007	0.333 ± 0.014	0.439 ± 0.020
LR	0.166 ± 0.004	0.343 ± 0.015	0.428 ± 0.018
MM	0.161 ± 0.007	0.330 ± 0.019	0.427 ± 0.020
ZN	0.159 ± 0.011	0.327 ± 0.022	0.432 ± 0.015

C = control, LG = L-glutamine (7 g/L), Q = quercetin (700 mg/L), SEB = slippery elm bark (210 mg/L), MR = marshmallow root (1340 mg/L), NAG = N-acetyl-D-glucosamine (210 mg/L), LR = licorice root (210 mg/L), MM = maitake mushrooms (42 mg/L), and ZN = zinc orotate (70 mg/L). There were no significant (*p* > 0.05) differences between treatments at a given time.

## Data Availability

Data is contained within the article.

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
