# Peer review of "Attributes of Culture Bacteria as Influenced by Ingredients That Help Treat Leaky Gut"

_microorganisms, 2023, doi:10.3390/microorganisms11040893_

Round 1

Reviewer 1 Report

The manuscript is well presented, however, I have some concern on the following;

a) Using a weak statistical method (LSD) to analyze the data.

b) Moderate English changes

Author Response

Reviewer 1

Comments and Suggestions for Authors

The manuscript is well presented, however, I have some concern on the following;

  1. Using a weak statistical method (LSD) to analyze the data.

RESPONSE: The statistical analysis is now well explained in the material and methods section. A table is now added to better describe the statistical analyses.

  1. Moderate English changes

RESPONSE: English changes were made to improved manuscript…

Reviewer 2 Report

In this research article, Aleman et al. tried to determine the influence of these ingredients on probiotic characteristics, tolerance to gastric juices and lysozyme, protease activity, and viability of Streptococcus thermophilus and Lactobacillus bulgaricus. 

I consider that the topic of research is relevant to the field, but there are limitations:

  • The authors published a similar study titled, ‘Probiotic Characteristics of Streptococcus thermophilus and Lactobacillus delbrueckii ssp. bulgaricus as Influenced by Carao (Cassia grandis)’ Fermentation 2022, 8(10), 499; https://doi.org/10.3390/fermentation8100499
  • But authors didn’t mention how this study differed from their previous work and also discussed their present results with their previous one.
  • There is no mechanism established with molecular-level evidence.
  • Figures are of poor quality; statistics need to be more accurate in figures.
  • The manuscript text needs to be better prepared, and there are still deletions in between in many places.

Author Response

Reviewer 2

Comments and Suggestions for Authors

In this research article, Aleman et al. tried to determine the influence of these ingredients on probiotic characteristics, tolerance to gastric juices and lysozyme, protease activity, and viability of Streptococcus thermophilus and Lactobacillus bulgaricus. 

I consider that the topic of research is relevant to the field, but there are limitations:

  • The authors published a similar study titled, ‘Probiotic Characteristics of Streptococcus thermophilus and Lactobacillus delbrueckii ssp. bulgaricus as Influenced by Carao (Cassia grandis)’ Fermentation 2022, 8(10), 499; https://doi.org/10.3390/fermentation8100499.  But authors didn’t mention how this study differed from their previous work and also discussed their present results with their previous one.

RESPONSE:    The earlier manuscript was on Carao only.  This manuscript is not on carao but includes includes 8 different ingredients that help in prevention of leaky gut which can lead to prevention of various diseases.  Furthermore, this manuscript also includes 2 additional characteristics “Tolerance to Simulated Gastric juices” and “Resistance to Lysozyme” not in the earlier manuscript.  Also, in the introduction, we have included how this study differed from their previous work and also discussed their present results with the previous Study.  Although both studies are distinctly different, the title can be confusing, so the title has now been changed to “Attributes of culture bacteria as influenced by ingredients that help treat leaky gut” to better reflect the current work.

  • There is no mechanism established with molecular-level evidence.

RESPONSE:    This was not the objective of the current study. 

  • Figures are of poor quality; statistics need to be more accurate in figures.

  • RESPONSE:    Since the figures display data of 8 ingredients + Control, there is a lot of information in the figures.  We tried to put in the statistics A, B, etc.. in the figures, but the letters overlapped and the figure looked crowded with information.  So, we just included an additional table for the statistics and explained the statistics of the figures in the text. 

  • The manuscript text needs to be better prepared, and there are still deletions in between in many places.

RESPONSE:    Manuscript is now better prepared and English changes were made in some spots. 

Reviewer 3 Report

In the introduction lists of various substances that have shown potential benefits through gut microbiota and intestinal barrier functions are listed in one sentence. I think it should be explained in more detail or at least classified according to their mechanism.

"S. thermophilus and L. bulgaricus are considered probiotics and used

it produces yogurt"

The probiotic property of the bacteria is not the same as using the bacteria as a starter culture. The sentence should be reformulated, because looking at these two bacteria as a species, we are talking about starter cultures that are used in the production of classic yogurt.

Figure 1 shows the acid tolerance of S. thermophilus under different treatments. It is not clear to me why you started the experiment with a different dose at the start. I think it is difficult to compare the curves if the starting dose is in the range of log 6 to 8.5. Can you normalize the result to see if there is a difference in the curves?

Figure 5 again shows different starting doses of bacteria. Can you explain that?

Can you comment on the large standard deviations and dispersion of the number of bacteria at the start of the experiment (0h)?

I think that in the propositions for sending manuscripts for review, it is written that tables and graphs should be incorporated into the text.

Author Response

Reviewer 3

In the introduction lists of various substances that have shown potential benefits through gut microbiota and intestinal barrier functions are listed in one sentence. I think it should be explained in more detail or at least classified according to their mechanism.

RESPONSE:    Explanation of the mechanism of action of these ingredients, is now included in the introduction.

"S. thermophilus and L. bulgaricus are considered probiotics and used it produces yogurt"

RESPONSE:    The terminology was changed

The probiotic property of the bacteria is not the same as using the bacteria as a starter culture. The sentence should be reformulated, because looking at these two bacteria as a species, we are talking about starter cultures that are used in the production of classic yogurt.

RESPONSE:    The terminology was changed

Figure 1 shows the acid tolerance of S. thermophilus under different treatments. It is not clear to me why you started the experiment with a different dose at the start. I think it is difficult to compare the curves if the starting dose is in the range of log 6 to 8.5. Can you normalize the result to see if there is a difference in the curves?

RESPONSE:    The Mean Log Differences are now reported in a new Table 3, which normalizes the starting does ranges.

Figure 5 again shows different starting doses of bacteria. Can you explain that?

RESPONSE:    The reviewer probably meant Figure 8 which had different starting doses.  Again, to normalize for the different starting doses the Mean Log Differences are now reported in the new Table 3.

Can you comment on the large standard deviations and dispersion of the number of bacteria at the start of the experiment (0h)?

RESPONSE:    The 0 h three replications were prepared from fresh samples on 3 different days (and are not 3 readings prepared 3 times from the same container).

I think that in the propositions for sending manuscripts for review, it is written that tables and graphs should be incorporated into the text.

RESPONSE:    We initially tried that, but then the Figure legends and parts of the tables spilled on to the next page and we thought that it would be easier to review if the Figures and their legends and the entire Tables were on the contained on their stand alone page.   

Thank you.

Round 2

Reviewer 2 Report

Thank you for submitting the revised manuscript by addressing my comments.

Reviewer 3 Report

I have no further comments.